# Preparation, Physicochemical Assessment and the Antimicrobial Action of Hydroxyapatite–Gelatin/Curcumin Nanofibrous Composites as a Dental Biomaterial

**DOI:** 10.3390/biomimetics7010004

**Published:** 2021-12-27

**Authors:** Simin Sharifi, Asma Zaheri Khosroshahi, Solmaz Maleki Dizaj, Yashar Rezaei

**Affiliations:** 1Department of Dental Biomaterials, Faculty of Dentistry, Tabriz University of Medical Sciences, Tabriz 51368, Iran; sharifi.ghazi@gmail.com (S.S.); asma.zaheri97@gmail.com (A.Z.K.); 2Dental and Periodontal Research Center, Tabriz University of Medical Sciences, Tabriz 51368, Iran

**Keywords:** nanocomposites, nanofibers, hydroxyapatite, gelatin, curcumin, antimicrobial effects, scaffold

## Abstract

In this study, we prepared and evaluated hydroxyapatite–gelatin/curcumin nanofibrous composites and determined their antimicrobial effects against *Escherichia coli*, *Staphylococcus aureus,* and *Streptococcus mutans*. Hydroxyapatite–gelatin/curcumin nanofibrous composites were prepared by the electrospinning method. The prepared nanocomposites were then subjected to physicochemical studies by the light scattering method for their particle size, Fourier transmission infrared spectroscopy (FTIR) to identify their functional groups, X-ray diffraction (XRD) to study their crystallinity, and scanning electron microscopy (SEM) to study their morphology. For the microbial evaluation of nanocomposites, the disk diffusion method was used against *Streptococcus mutans*, *Staphylococcus aureus*, and *Escherichia coli*. The results showed that the nanofibers were uniform in shape without any bead (structural defects). The release pattern of curcumin from the nanocomposite was a two-stage release, 60% of which was released in the first two days and the rest being slowly released until the 14th day. The results of the microbial evaluations showed that the nanocomposites had significant antimicrobial effects against all bacteria (*p* = 0.0086). It seems that these nanocomposites can be used in dental tissue engineering or as other dental materials. Also, according to the appropriate microbial results, these plant antimicrobials can be used instead of chemical antimicrobials, or along with them, to reduce bacterial resistance.

## 1. Introduction

Antibiotic-resistant microorganisms have prompted researchers to explore nano-optimization for drug delivery to specific regions [1,2,3]. The surface-to-volume proportion of nanoscale particles has risen dramatically due to their manufacturing, increasing their efficacy even at minimal concentrations [4].

A variety of beneficial effects of herbal medicines have been identified [5,6,7,8]. Curcumin (diferuloylmethane) is achieved from the *Curcuma longa* (Turmeric) rhizome. Lately, it has received important attention as a medicinal plant owing to its distinctive therapeutic benefits including antimicrobial, anti-pathogenic, antioxidant and anti-inflammatory, anti-angiogenic, anticancer, and anti-diabetic effects [9,10,11,12]. Despite its outstanding biological effects, there are limitations to its clinical use owing to its low bioavailability. Recently, several nanoparticles such as nanofibers, solid lipid nanoparticles, nanostructured lipid carriers, liposomes, micelles, nanogels, and magnetic nanoparticles have advanced as potential strategies to improve the therapeutic effects of curcumin [9,10,11,12,13,14,15]. Different reports have confirmed that the novel systems of curcumin and stem cell differentiation have great therapeutic potential against various bone-related diseases and disorders [16].

There are many dental uses for composites. Nanocomposites have recently been used enormously in the dentistry field. For the treatment of oral disorders, they may be utilized as drug delivery systems, fillers, restorative substances, and tissue engineering components [17,18,19]. Fiber nanocomposites have a high surface-to-volume ratio, owing to their fiber shape. Studies emphasize applying localized antibiotic-containing three-dimensional nanofibers in combination with stem-cell-enriched injectable scaffolds or growth factors and an intracanal medication delivery approach. Dental pulp regeneration in humans may be improved using nanofibers [20,21]. Several medicines may be carried via nanofibers. Antibiotics, anticancer drugs, DNA, RNA, proteins, and growth factors may be delivered through polymer nanofibers [21]. 

Composite fiber scaffolds can create an environment that can facilitate adhesion, proliferation, and cell differentiation. Nanofibers are used to mimic a variety of tissues by mimicking the extracellular matrix with a porous network. Nanofiber scaffolds not only enhance the regeneration of dental tissues, but can also advance the technology for tissue engineering replacements in many physiological systems. Besides, this technology can assist in maxillofacial surgeries to facilitate surgical procedures and to reduce their costs [22,23]. 

The electrospinning technique is the most common way to make fibers in the laboratory and on a large scale in the industry [9,24]. The electric jet force used in electrospinning yields polymer nanofibers. It is possible to create more sophisticated nanofibers by mixing different polymer solutions with various other elements, such as pharmaceuticals, other nanoparticles, and even cells. Using these particular nanofibers as biocompatible solutions or gels has many therapeutic advantages [25]. Antibiotic-containing nanofibers, for instance, have been studied for their impact on bacterial biofilm formation. The microscopic scans of infected dentin subjected to these nanofibers revealed significant microbial mortality. Numerous investigations on the efficacy of these nanofibers throughout animals with periapical illnesses have been published [26,27,28]. 

Enamel, dentin, cementum, and bone include hydroxyapatite, a crystalline calcium phosphate [29]. Because of its tooth-like microstructure, nanohydroxyapatite is effective in remineralizing teeth and restoring their fracture resistance. In addition to its ability to decrease tooth susceptibility, clinical and experimental investigations have revealed that this compound may also inhibit tubules [30]. 

Gelatin is a kind of amorphous collagen that is made by gradually melting Type I collagen. It has the benefit of being less expensive and less complicated to produce when compared to collagen. Additionally, hydroxyapatite–gelatin is more effective than hydroxyapatite–collagen in triggering osteoblast reactions [31]. Composites made of gelatin and hydroxyapatite have recently been suggested as drug delivery vehicles to bone cells [32,33]. 

In this study, we prepared and evaluated hydroxyapatite-gelatin/curcumin fiber nanocomposites and determined their antimicrobial effects against *Escherichia coli*, *Staphylococcus aureus,* and *Streptococcus mutans*.

## 2. Materials

Curcumin, gelatin, 2, 2, 2 trifluoroethanol, and dimethyl sulfone were provided from Sigma Aldrich (St. Louis, MO, USA). From the Nanobazar Company (Tehran, Iran), hydroxyapatite nanoparticles with an average particle size of 40 nm were purchased. Muller-Hinton agar was purchased from Thermo Fisher Company (Melbourne, Australia).

## 3. Methods

### 3.1. Nanocomposites Fabrication 

Nanocomposites comprising gelatin, hydroxyapatite, and curcumin fiber were made using an electrospinning apparatus (Nanofanavaran, Mashhad, Iran) in a vertical electrospinning process with a fixed collector. There was a 75:25 proportion of gelatin to curcumin in the organic gelatin–curcumin solution made in 2,2,2 trifluoroethanol (yellow solution). Additionally, hydroxyapatite nanoparticles were suspended in a solution containing 8% gelatin (white solution). Five-milliliter syringes were filled with yellow and white solutions, respectively. A micrometer syringe head was used to connect a nozzle to the tips of the syringes before they were put into the electrospinning apparatus. Aluminum foil was used to wrap the device’s collection plate. A few crucial device variables were set as follows: 20 kV voltage, a spacing of 10 cm between the syringe nozzle tip and the collection plate, and a flow rate of 1.5 mL/h for a solution coming out of the nozzle. When the device’s starting button was pressed, the required solutions were hurled from the nozzle’s tip to an aluminum foil-covered spinning collection plate on the other end. Throughout the fast launch procedure, the organic solvent vaporized. Nanometer fibers, the ultimate product, were gathered on the plate. The collection plate and aluminum foil were carefully separated. Drying the fibers at room temperature was carried out. The sample was placed in a refrigerator at −18 °C. To examine the antibacterial properties of curcumin, gelatin nanofibers comprising hydroxyapatite nanoparticles free of curcumin were produced in the same manner. In addition, gelatin–curcumin nanofibers free of hydroxyapatite nanoparticles were manufactured to similarly examine the resulting composite’s characteristics. 

### 3.2. Nanoparticle’s Particle Size

Dynamic light scattering (DLS, Malvern, Cambridge, Massachusetts, UK) was used to verify the nanoscale size precision of the produced nanofibers at 25 °C, using an argon laser beam at 633 nm and a 90° scattering angle. For this, distilled water was used to make a high-quality suspension of nanofibers, which was then put into the device’s specific tube. 

### 3.3. Morphology of Nanoparticles

Electron microscopy was used to examine the nanoparticles’ morphological aspects (Razi Company of Tehran, Tehran, Iran). Using a scanning electron microscope (SEM, TESCAN-USA) under a high-vacuum atmosphere and at an acceleration voltage of 10 kV, the powdered specimens were put on an SEM plate and then gold-coated. The SEM magnification was selected to be 50 kx.

### 3.4. The Loading Efficiency and the Release Pattern of Curcumin

Ten milligrams of nanofibers were dissolved in an organic solvent (dimethyl sulfonide) to measure the quantity of curcumin loaded onto the nanofibers. An ultraviolet spectrophotometer was used to measure the UV absorbance. One milliliter of the dispersed nanofiber solution was put into a tube to obtain the adsorption number for curcumin, by setting the device’s λmax at 350 nm. 

The drug dissolution device No. 2 (Apparatus 2) was utilized to assess the curcumin release from nanofibers. Each of the six valves received 300 mL of the phosphate-buffered solution. Five mg of the nanofibers were added to each valve of the apparatus. The pH was set to 7.4. The temperature was adjusted to 37 °C. The speed was adjusted to 100 rpm. Daily, one milliliter of the specimen was collected from the valves, and the UV absorption was determined employing a spectrophotometer. One milliliter of fresh buffer medium was used to substitute the specimen collected. 

### 3.5. X-ray Diffraction (XRD) Analysis

The materials were analyzed using XRD patterns at room temperature. An X-ray diffractometer (D5000, Siemens, Munich, Bavaria, Germany), a wavelength of 5405/1 Å, a voltage of 40 kV, and a current of 30 mA were used to measure the patterns of the samples. Ultimately, their patterns were determined in a range from zero to sixty 2-theta degrees.

### 3.6. Fourier Transform Infrared Spectroscopy (FTIR)

The functional groups were identified using Fourier transmission infrared spectroscopy (FTIR, Shimadzu 8400S-Japan, Kyoto, Japan). The samples were mixed with potassium bromide of IR grade and compressed via an IR pellet manufacturing machine. Then, the wavelengths were set from 400 to 4000 (cm^−1^). 

### 3.7. Microbial Methods

*Staphylococcus aureus* (ATCC: 6538), *Escherichia coli* (ATCC: 25922), and *Streptococcus mutans* (ATCC: 25175) bacteria were provided from the Pasteur Institute of Iran (Tehran, Iran). A disk diffusion procedure was applied to examine the produced nanocomposite’s antimicrobial performance. Vancomycin (30 mg per disc) and rifampicin (5 mg per disc) were used as positive controls. Antibacterial actions of the prepared fibers were assessed by a disk diffusion agar on the Muller-Hinton agar. Briefly, the bacterial suspension, equivalent to the 0.5 McFarland standard (1.5 × 108 CFU/mL), was inoculated on the Muller-Hinton plates using a swab and allowed to dry for 10 min. Six millimeter disks of samples were located on the agar surface and plates were incubated at 35 °C for 24 h. Finally, the inhibition zones around the disks were measured.

### 3.8. Statistical Analysis

The results were reported as mean ± SD and frequency (percentage). Data normality was assessed using the Kolmogorov–Smirnov test. To compare the findings of the average inhibition zone across microorganisms and among the investigated groups, a one-way ANOVA was employed. SPSS software (version 16, IBM, New York, NY, USA) was used to analyze the data. The *p*-values of less than 0.05 were considered as the significance level. 

## 4. Results

### 4.1. Mean Particle Size

The manufactured nanofibers’ mean particle size is shown in Figure 1. The findings demonstrated that the mean particle size was 98 nanometers.

### 4.2. Morphological Assessment

Images of scanning electron microscopy (SEM) for gelatin–curcumin nanofibers and hydroxyapatite–gelatin/curcumin nanocomposites are shown in Figure 2. The results show that the nanofibers are uniform in shape without any bead (structural defect). Their dimensions are measured in nanometers. Nanocomposite images show the presence of hydroxyapatite nanoparticles on the fibers as well.

### 4.3. Release Pattern of Curcumin

The release of curcumin from hydroxyapatite–gelatin/curcumin nanocomposites is shown in Figure 3. Curcumin was released from the nanofibers in a two-step sequence, 60% in the initial two days and the rest gradually over the next 14 days. 

### 4.4. X-ray Diffraction (XRD) Analysis

Hydroxyapatite–gelatin/curcumin nanocomposites and their purified forms are illustrated in Figure 4 in terms of their X-ray diffraction patterns.

### 4.5. Fourier Transform Infrared Spectroscopy (FTIR) Analysis

Figure 5 depicts the Fourier transmission infrared spectroscopy (FTIR) findings. FTIR results revealed no novel interactions among the investigated compounds.

### 4.6. Microbial Findings

All three tested microorganisms showed inhibition zones in the presence of the nanocomposite, as shown in Figure 6a–c. No inhibitory outcomes were obtained using gelatin–hydroxyapatite nanofibers without curcumin.

### 4.7. Statistical Analysis for Microbial Findings

Table 1 illustrates the inhibition zone for the produced substances and their positive and negative controls against *S. mutans*, *S. aureus*, and *E. coli*.

Table 2 displays the conclusions of the one-way ANOVA for the produced nanocomposite (analysis of the inhibition zone between these three microbes). *Staphylococcus aureus* exhibited the greatest inhibition zone, followed by *Escherichia coli* and *Streptococcus mutans*. For the synthesized nanocomposite, there was a significant disparity in the extent of inhibition zones among the bacteria (*p* = 0.0086).

## 5. Discussion

It is necessary to specify the physicochemical characteristics of nanocarriers before using them for various purposes, to guarantee their appropriateness. The SEM image revealed that the nanofibers were arranged in a lattice-like pattern. Their measurements matched up with a nanometer-scale microscopic assessment. No structural defects were found in the nanofibers based on the taken images. This composite was effectively produced when the images of gelatin–curcumin nanofibers were compared to the gelatin–curcumin–hydroxyapatite nanocomposite. When compared to the other scaffolds in vivo and in vitro, electrospun nanofiber scaffolds have shown an outstanding capacity to direct cell motility, morphological aspects of cell shape and eventually influence cell differentiation [34,35,36]. The investigations have also demonstrated the possibility of cell compatibility throughout this fibrous layer as a nanoscale microenvironment, due to particular biological processes, including cell differentiation, attachment, and motility [37].

Our findings indicated that curcumin was released from the produced nanocomposite in a two-step mechanism that began quickly (60%) on the first and second days and, after that, was then gradually released. A drug may be delivered from a biodegradable matrix through various processes, namely through molecular diffusion from the matrix, matrix breakdown over time, and material degradation, or a combination of both processes [38]. Therefore, it appears that the diffusion process is responsible for the first immediate release of curcumin from the nanofiber matrix in this experiment. The sustained release is then due to the degradation of the nanofiber matrix [38,39]. Shabdoost et al. demonstrated that curcumin was delivered in two phases from the polyurethane nanofiber matrix. They found that the first, fast release lasted one day, and the subsequent, gradual release persevered for an additional eleven days [39]. According to the study by Boroumand et al., fifty-eight percent of curcumin was released from the polycaprolactone nanofiber matrix during the first day. The residual percent was then released after 30 days [38]. 

X-ray diffraction pattern findings revealed that all purified materials in our research (hydroxyapatite with 26, 31, 39, 41 2-theta and curcumin with 8, 9, 12, 14, 17 2-theta) exhibited their index peaks [40]. All components’ peaks may be observed in the produced nanocomposite’s peak, as well. The explanation for the reduction in peak intensity throughout the nanocomposites is the transformation of substances to the nanoscale, as well as the amorphous structure of gelatin, which, as the largest proportion of the matrix, pushes the peak primarily towards the amorphous region. Polylactic acid–hydroxyapatite–curcumin nanocomposites studied by Hazma et al. exhibited comparable outcomes [40]. The ester, the ketone, and the ether groups of curcumin had absorption values of 1720, 1650, and 11,300 cm^−1^, respectively, according to the findings of FTIR spectroscopy. It has also been shown that the broad peak at 3200–3500 cm^−1^ is linked to the OH group’s tension spectrum [41]. At 563 and 602 cm^−1^, the index peaks are associated with the hydroxyapatite phosphate group [40], and at 1652 cm^−1^, the amide bands associated with gelatin may be observed [42].

*Staphylococcus aureus* exhibited the greatest inhibition zone, followed by *Escherichia coli* and *Streptococcus mutans*. For the synthesized nanocomposite, there was a significant difference in the extent of the inhibition zone among the bacteria (*p* = 0.0086). There were comparable antibacterial findings against *Staphylococcus aureus* and *Escherichia coli* when Ghavimi et al. investigated collagen–curcumin nanofibers’ antimicrobial capabilities [9]. Curcumin has shown to have comparable impacts on Gram-positive and Gram-negative bacteria in other studies [43,44,45]. 

Nanomaterials apply their antibacterial actions on bacteria by several mechanisms [46,47]. Based on the reports, the physicochemical possessions of nanoparticles are actually vital in their antimicrobial properties [48]. Nanomaterials have the ability to disrupt cell membrane functions by binding to the surface of cell membranes with a high affinity. This effect is more predominant in smaller nanoparticles, owing to their larger surface space [49,50,51]. The interaction between the membrane and nanomaterials also leads to local pores in the membrane and harms the bacteria due to the passing of nanoparticles into the bacteria and the interaction of bacteria‘s proteins with DNA [52]. Some nanomaterials are also able to mix with the bacterial cell wall to release their antimicrobial material into the cytoplasm [46,53]. Antimicrobial nanofibers can also pass through the bacteria pores due to their very small diameter and can then disrupt the bacteria’s different functions [54,55].

### Conclusions and the Future Perspectives

The results showed that the hydroxyapatite–gelatin/curcumin nanocomposites were uniform in shape without any structural defects and had a two-stage release pattern of curcumin from nanocomposite. Besides, the prepared nanocomposites had significant antimicrobial effects against all bacteria. The antimicrobial activity of curcumin against the selected bacteria showed that herbal antimicrobials may be used as a substitute for chemical anti-bacterials in the future to decrease bacterial resistance. Moreover, the prepared nanocomposites may create an environment that can facilitate adhesion, proliferation, and cell differentiation by mimicking the extracellular matrix. They can enhance the regeneration of dental tissues to advanced levels. For example, they may assist maxillofacial surgeries to facilitate surgical procedures and to reduce their costs. The cellular studies are recommended to determine if hydroxyapatite nanoparticles induce bone tissue effects. Curcumin-induced cell proliferation and tissue healing may be studied in vitro as well. Also, further animal and human studies are required for the establishment of the actual therapeutic usefulness of this novel substance.

## 6. Ethical Considerations

The ethics committee at Tabriz University of Medical Sciences provided the ethical code, and all procedures were carried out after getting the ethical code (IR.TBZMED.VCR.REC.1399.140). 

## Figures and Tables

**Figure 1 biomimetics-07-00004-f001:**
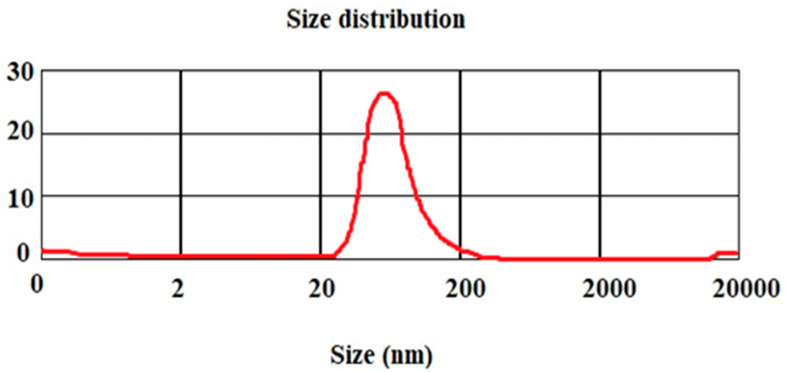
Mean particle size for hydroxyapatite–gelatin/curcumin nanocomposites.

**Figure 2 biomimetics-07-00004-f002:**
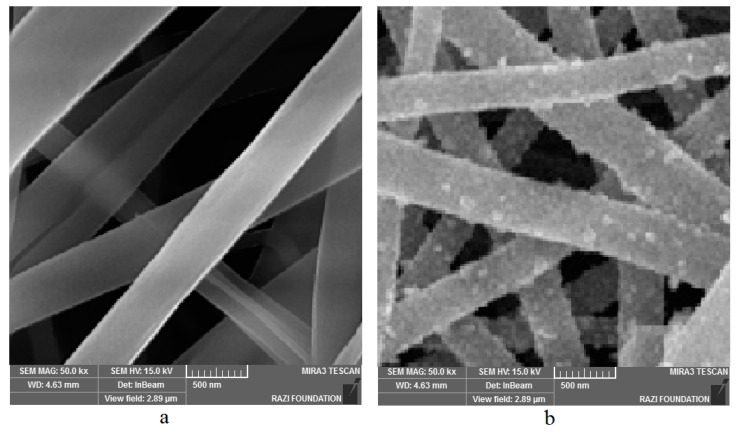
Images of scanning electron microscopy (SEM): gelatin–curcumin nanofibers (**a**) and hydroxyapatite–gelatin/curcumin nanocomposites (**b**).

**Figure 3 biomimetics-07-00004-f003:**
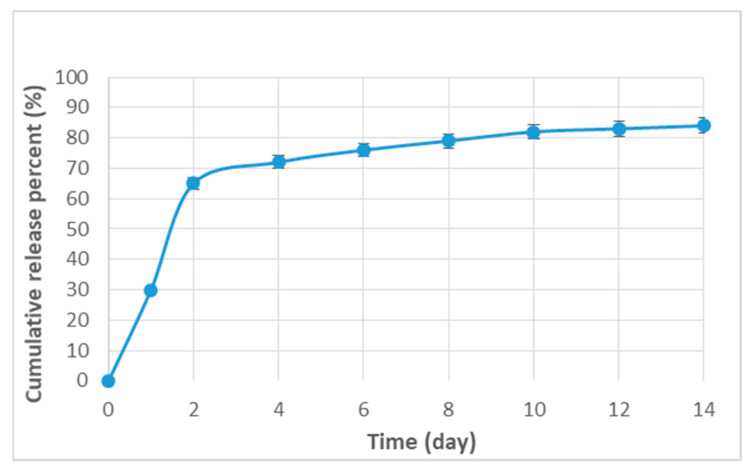
The release pattern of curcumin from hydroxyapatite–gelatin/curcumin nanocomposites.

**Figure 4 biomimetics-07-00004-f004:**
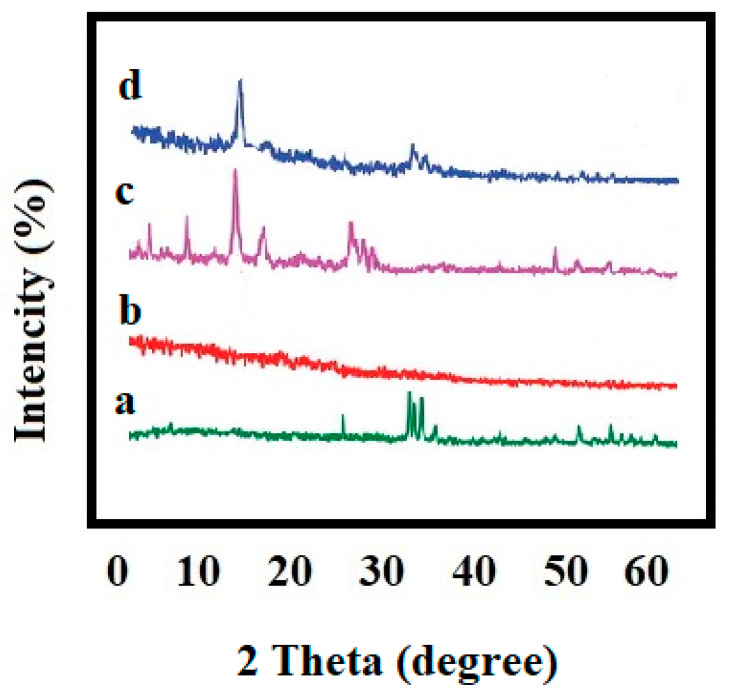
The pattern of X-ray diffraction for nanoparticles of hydroxyapatite (**a**), gelatin (**b**), curcumin (**c**), and hydroxyapatite–gelatin/curcumin nanocomposites (**d**).

**Figure 5 biomimetics-07-00004-f005:**
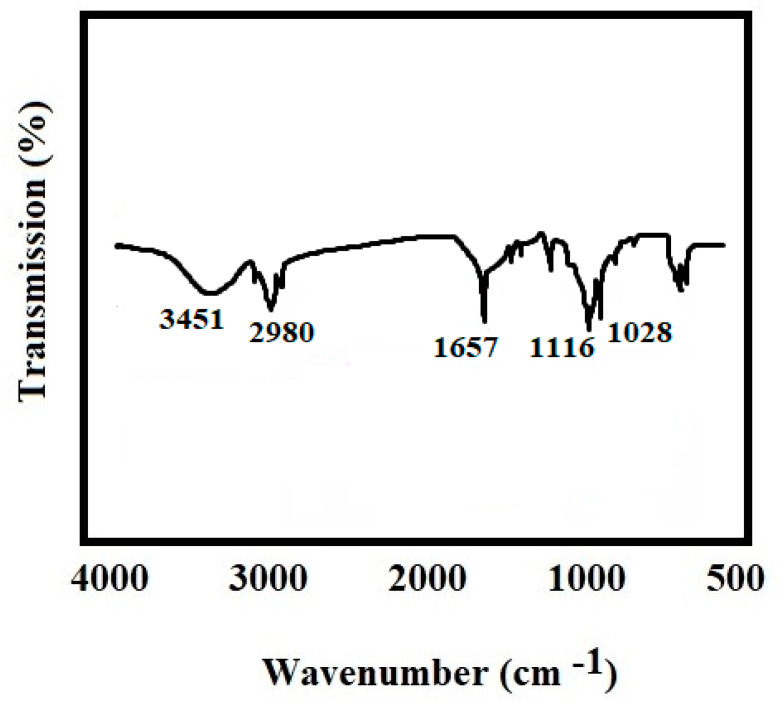
An FTIR spectrum for hydroxyapatite–gelatin/curcumin nanocomposites.

**Figure 6 biomimetics-07-00004-f006:**
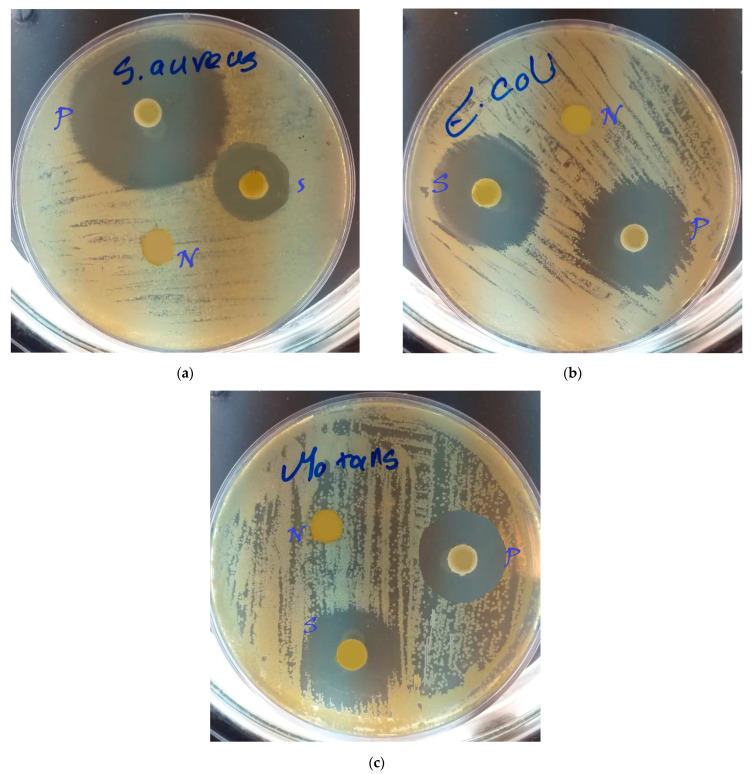
Findings of the antimicrobial assessment for three types of bacteria; (**a**) *Staphylococcus aureus*, (**b**) *Escherichia coli* and (**c**) *Streptococcus mutans*.

**Table 1 biomimetics-07-00004-t001:** Sizes of inhibition zones for the manufactured substances, positive and negative controls.

	Zone Size of Sample	Zone Size of Positive Control	Zone Size of Negative Control
*S. mutans*	13.17 ± 0.05	12.6 ± 0.03	0
*E. coli*	14.43 ± 0.01	14.1 ± 0.08	0
*S. aureus*	10.7 ± 0.05	16.13 ± 0.02	0

**Table 2 biomimetics-07-00004-t002:** One-way ANOVA findings for synthesized nanocomposites (comparison of inhibition zone between three microbes).

ANOVA Summary	
F	115.1
*p* value	0.0086
*p* value summary	**
Significant diff. among means (*p* < 0.05)?	Yes

## Data Availability

The raw/processed data needed to reproduce these outcomes can be shared at this time. Also, after publication, the data can be requested from corresponding author via email.

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
