# Peer review of "Preparation, Physicochemical Assessment and the Antimicrobial Action of Hydroxyapatite–Gelatin/Curcumin Nanofibrous Composites as a Dental Biomaterial"

_biomimetics, 2021, doi:10.3390/biomimetics7010004_

Round 1

Reviewer 1 Report

Comments and Suggestions for Authors

In this work, the authors have prepared and evaluated hydroxyapatite-gelatin/curcumin nanofibrous composites and determined their antimicrobial effects against Escherichia coli, Staphylococcus aureus and Streptococcus mutans. The nanofibrous composites studied were obtained by electrospinning procedure, and the interpretation of data is reasonable. However, a minor revision is necessary before publication.

Some suggestions / comments:

  1. Some small spelling mistakes appear in the text. Increasing the quality of the English language would be necessary.
  2. Some details about the electrospinning procedure would be necessary, eg: Horizontal or vertical electrospinning process? Fixed or rotary collector? If it is rotary, at what rotational speed was it worked?
  3. The authors should discuss something more about the novelty of the product obtained in dentistry. What are the benefits compared to other products?
  4. Check references and then write in accordance with the journal's instructions.
  5. In the text, reference numbers should be placed in square brackets [ ], and placed before the punctuation; for example [1], [1–3] or [1,3].

Author Response

Thanks for your valuable comments. I attached the responses.

Reviewer 2 Report

This study aims to prepare and characterize hydroxyapatite-gelatin/curcumin nanofibrous composites with antimicrobial effect for dental use. Suitable methods like FTIR, SEM, XRD are used to fulfill this task. The release pattern of curcumin was also described.
However, most methods are not well described. E.g., SEM, FTIR, antimicrobial methods, statistical analysis are described very poorly, and they should be described using software, details. Especially antimicrobials methods should be described more clearly, in more detail, a reference for standard method should be included. The supplier of agar media is not specified. Is it true that disks were put on the medium after 24h of bacterial cultivation? The principle of the disk diffusion method is that bacteria are inoculated on Muller Hinton agar and not more than till 15 minutes should be disks put on. And now the cultivation can be done.

Results are described very poorly, and the quality of figures 1, 2, 4, 5 is deficient.

Discussion should be broadened. Surface interactions could be discussed with literature. The oral microbiome is the second mostly inhabited place on the human body. There is a wide range of different bacteria, especially anaerobes. Antimicrobial tests should be done at least against some of them.

The conclusion should be more clear and precise about what was found by presented experiments.

More than half references are older than 5 years.

Author Response

(The authors gave the same response as above.)

Round 2

Reviewer 2 Report

Authors made the requested imrpovements and manuscript could be published. Až